# Comparative Genomics of *Bryopsis hypnoides*: Structural Conservation and Gene Transfer Between Chloroplast and Mitochondrial Genomes

**DOI:** 10.3390/biom15020278

**Published:** 2025-02-13

**Authors:** Ziwen Liu, Xiao Fan, Yukun Wu, Wei Zhang, Xiaowen Zhang, Dong Xu, Yitao Wang, Ke Sun, Wei Wang, Naihao Ye

**Affiliations:** 1State Key Laboratory of Mariculture Biobreeding and Sustainable Goods, Yellow Sea Fisheries Research Institute, Chinese Academy of Fishery Sciences, Qingdao 266071, China; liuziwen9903@gmail.com (Z.L.); fanxiao@ysfri.ac.cn (X.F.); wuyukun@stu.ouc.edu.cn (Y.W.); zhangwei9007@stu.ouc.edu.cn (W.Z.); zhangxw@ysfri.ac.cn (X.Z.); xudong@ysfri.ac.cn (D.X.); wangyt@ysfri.ac.cn (Y.W.); sunke@ysfri.ac.cn (K.S.); wangwei@ysfri.ac.cn (W.W.); 2Graduate School of Chinese Academy of Agricultural Sciences, Beijing 100081, China; 3Laboratory for Marine Fisheries Science and Food Production Processes, Laoshan Laboratory, Qingdao 266237, China

**Keywords:** chloroplast genome, mitochondrial genome, gene transfer

## Abstract

*Bryopsis hypnoides*, a unicellular multinucleate green alga in the genus *Bryopsis*, plays vital ecological roles and represents a key evolutionary link between unicellular and multicellular algae. However, its weak genetic baseline data have constrained the progress of evolutionary research. In this study, we successfully assembled and annotated the complete circular chloroplast and mitochondrial genomes of *B. hypnoides*. The chloroplast genome has a total length of 139,745 bp and contains 59 protein-coding genes, 2 rRNA genes, and 11 tRNA genes, with 31 genes associated with photosynthesis. The mitochondrial genome has a total length of 408,555 bp and contains 41 protein-coding genes, 3 rRNA genes, and 18 tRNA genes, with 18 genes involved in oxidative phosphorylation. Based on the data, we conducted a genetic comparison involving repeat sequences, phylogenetic relationships, codon usage preferences, and gene transfer between the two organellar genomes. The major results highlighted that (1) the chloroplast genome favors A/T repeats, whereas the mitochondrial genome prefers C/G repeats; (2) codon usage preference analysis indicated that both organellar genomes prefer codons ending in A/T, with a stronger bias observed in the chloroplast genome; and (3) sixteen fragments with high sequence identity were identified between the two organellar genomes, indicating potential gene transfer. These findings provide critical insights into the organellar genome characteristics and evolution of *B. hypnoides*.

## 1. Introduction

*Bryopsis hypnoides*, a unicellular and multinucleate green alga of the genus *Bryopsis*, is widely distributed from tropical to polar oceans, typically growing on intertidal and subtidal rocks or in rock pools. Its unique unicellular and multinucleate structure may represent a transitional stage in the evolution from unicellular to multicellular organisms, raising many genetic questions [1,2]. As such, *B. hypnoides* provides new perspectives and clues for understanding the complex mechanisms of genetic evolution. Moreover, protoplast contains all the essential components for cellular life activities and serves as an excellent model for exploring genetic mechanisms’ underlying evolution. In laboratory conditions, the extruded cytoplasm of *B. hypnoides* rapidly aggregates and transforms into protoplast [3,4,5,6], garnering significant scientific interest. However, detailed studies on this species remain scarce, limiting comprehensive insights into its genetic and evolutionary mechanisms.

Chloroplasts and mitochondria are the only organelles with endosymbiotic-origin genomes [7], regulating key physiological processes, such as photosynthesis and respiration, and playing essential roles in growth and development. Their genomes retain only a few functional genes [8,9], as many have been transferred to the nuclear genome during evolution [10]. Investigating these organellar genomes not only reveals their functional mechanisms, but also sheds light on genomic rearrangements, gene loss, and transfer during different evolutionary stages, providing robust evidence for understanding the complexity and diversity of the evolution of life.

Within a single species, chloroplast and mitochondrial genomes exhibit significant differences, including in terms of structural organization, size, expression patterns, and gene composition, largely influenced by the extent of gene transfer to and replacement by the nuclear genome. They also differ in mutation rates and structural variations. For instance, the evolutionary rate of the chloroplast genome is half that of the nuclear genome, while that of plant mitochondrial genomes is less than one-sixth that of the nuclear genome [11,12], correlating with adaptive evolution [13]. These differences manifest in the types, lengths, and numbers of repeat sequences and codon usage preferences, and may also lead to discrepancies in phylogenetic relationships inferred from chloroplast and mitochondrial data. Thus, a comparative analysis of these organellar genomes not only highlights their functional divergence and complementarity, but also helps uncover the mechanisms driving their evolution. However, due to the complexity of plant mitochondrial structures and the lack of specialized assembly tools, obtaining accurate and complete mitochondrial assembled genomes remains challenging. Consequently, we possess data for few species with both chloroplast and mitochondrial genomes, hindering further research in this field.

In addition, a complex gene transfer occurs between chloroplasts and mitochondria. This transfer involves not only intra-organellar recombination but also inter-organellar and organelle-to-nucleus gene transfer, known as “nuclear-organellar gene flow”. This phenomenon reveals the dynamics and continuity of the evolution of life, providing insights into how organisms adapt to environmental changes and optimize physiological functions. In *B. hypnoides*, investigating chloroplast–mitochondrial gene transfer can uncover its unique evolutionary history and serve as a valuable model for understanding the co-evolution of organelles and the nuclear genome.

In this study, we sequenced and assembled the complete chloroplast and mitochondrial genomes of *B. hypnoides*. We comprehensively analyzed the organellar genomes, examined their differences, and explored gene transfer between them. These findings aim to provide insights into the patterns and mechanisms of intracellular gene transfer across broader taxonomic levels, contributing to the advancement of organellar genomics and fostering deeper research in the field.

## 2. Materials and Methods

### 2.1. Sample Collection and Preservation

Specimens of *B. hypnoides* were collected from Qingdao, Shandong Province, China (120.33° E, 36.07° N) and are currently preserved at the Yellow Sea Fisheries Research Institute, Chinese Academy of Fishery Sciences (Qingdao, China). Fresh samples were washed with sterile water, air-dried, wrapped in aluminum foil, rapidly frozen in liquid nitrogen, and stored at −80 °C until further use.

### 2.2. Genomic DNA Extraction and Sequencing

A 0.1 g sample of *B. hypnoides* was ground in a liquid nitrogen-cooled mortar, and total genomic DNA was extracted using the CTAB method [14]. The quality and concentration of DNA were assessed using 1% agarose gel electrophoresis and a multifunctional microplate reader. Sequencing was performed on Illumina and PacBio platforms, generating raw sequencing data.

### 2.3. Genome Assembly and Annotation

The sequencing reads for mitochondrial and chloroplast assembly were extracted from whole genome data by referencing a library based on published organellar genomes. Chloroplast and mitochondrial genomes were assembled using Flye v2.9.2 [15] with coverage depths of 1933× and 1551×, respectively (typically, a coverage depth of 40× is enough [15]). The chloroplast genome of *B. hypnoides* (GenBank: GQ892829.1) and the mitochondrial genome of *Bryopsis* sp. KO-2023 (GenBank: LC768902.1) were used as reference genomes. Genome annotation was conducted using CPGAVAS2 (http://47.96.249.172:16019/analyzer/home, accessed on 25 November 2024), GeSeq [16] (https://chlorobox.mpimp-golm.mpg.de/geseq.html, accessed on 25 November 2024), and Geneious v9.0.2 [17], followed by manual curation and validation using GB2sequin [18] (https://chlorobox.mpimp-golm.mpg.de/GenBank2Sequin.html, accessed on 26 November 2024). Genome maps were generated with Chloroplot [19] (https://irscope.shinyapps.io/Chloroplot/, accessed on 17 December 2024).

### 2.4. Genome Validation

To ensure the accuracy of genome assembly, we employed Samtools v1.21 [20] to map sequencing reads back to the assembled genome and calculate the depth at each site. Visualization of the mapping results was conducted using Tablet v1.21.02.08 [21] to inspect depth coverage and identify any gap regions. We validated the absence of gaps in the assembled organellar genomes through two methods. Firstly, we manually inspected every locus across the entire genome in Tablet to verify the presence of overlaps of reads at each site. Secondly, we visualized the depth at each genomic locus and examined areas with exceptionally low depth to confirm the absence of gaps. Additionally, sequence alignments of the assembled genomes and reference genomes were conducted using MEGA11 [22], and PCR primers were designed for 14 loci (7 chloroplast loci and 7 mitochondrial loci) (Appendix A) to validate discrepancies through experimental verification. After multi-method validation, the complete chloroplast and mitochondrial genomes of *B. hypnoides* were submitted to GenBank (accession number: PQ652170 and PQ766611).

### 2.5. Comparative Genomics

Chloroplast genomes from 13 species (10 closely related species and 3 outgroups) and mitochondrial genomes from 8 species (5 closely related species and 3 outgroups) were retrieved from NCBI for comparative genomic analysis. Simple sequence repeats (SSRs) were identified using the MISA web tool [23,24] (https://webblast.ipk-gatersleben.de/misa/, accessed on 18 December 2024) with minimum repeat thresholds of 10, 6, 5, 5, 5, and 5 for mono-, di-, tri-, tetra-, penta-, and hexanucleotide motifs, respectively. Dispersed repeats (≥55 bp), including forward (F), reverse (R), complementary (C), and palindromic (P) repeats, were identified using REPuter [25] (https://bibiserv.cebitec.uni-bielefeld.de/reputer, accessed on 21 December 2024) (Maximum Computed Repeats 5000, Minimal Repeat Size 55 bp). Subsequently, we manually filtered out the results with e-values exceeding 1 × 10^−5^ and conducted statistical analyses on the filtered results, categorizing them according to repeat type (F, R, C, and P) and repeat length (55–60, 60–65, 65–70, 70–75, 75–80, 80–85, 85–90, 90–95, 95–100, 100–105, 105–110, >=110) to compare the differences in dispersed repeats among various species.

### 2.6. Phylogenetic Analysis

Phylogenetic trees were constructed using conserved genes shared by *B. hypnoides* and related species. For the chloroplast genome, *Cephaleuros diffusus*, *Cephaleuros virescens*, and *Cephaleuros lagerheimii* were used as the outgroup. For the mitochondrial genome, *Ulva flexuosa*, *Ulva prolifera*, and *Ulva compressa* were used as the outgroup. Multiple sequence alignments were performed using MAFFT v7.505 [26], and maximum likelihood (ML) trees were constructed with IQtree v2.0.3 [27] (-bb 1000-nt AUTO-m MFP). Neighbor-Joining (NJ) and Bayesian (Mrbayes) analyses were also conducted with bootstrap values set to 1000.

### 2.7. Synteny and Codon Usage Analysis

To investigate the synteny of protein-coding sequences between organellar genomes of *B. hypnoides* and other species, for each pair of adjacent species on the phylogenetic tree, we first conducted sequence alignment using BLASTP (-evalue 1 × 10^−5^-outfmt 6-num_alignments 5). Subsequently, we applied MCscanX [28] to detect collinear blocks using default parameters (MATCH_SCORE 50, MATCH_SIZE 5, GAP_PENALTY-1, OVERLAP_WINDOW 5, E_VALUE 1 × 10^−5^, MAX GAPS 25). Given the format discrepancy between the standard general feature format (GFF) files and the specific format required by MCscanX, we utilized a custom script to convert each species’ standard GFF files into the compatible format. Thereafter, conserved blocks identified using MCscanX were visualized and annotated. Genome annotation maps were drawn using OGDRAW [29] (https://chlorobox.mpimp-golm.mpg.de/OGDraw.html, accessed on 21 December 2024) with manual corrections. Common and unique gene distributions among species were visualized using ImageGP 2 [30] (https://www.bic.ac.cn/BIC/#/, accessed on 22 December 2024). Relative synonymous codon usage (RSCU) of chloroplast and mitochondrial genomes was calculated using CodonW v1.4.4 (http://codonw.sourceforge.net, accessed on 22 December 2024).

### 2.8. Organellar Genome Gene Transfer Analysis

To identify DNA fragment transfers between chloroplast and mitochondrial genomes, BLASTN (E ≤ 1 × 10^−5^) was used to perform sequence similarity analyses. Homologous regions with high similarity were identified and visualized to elucidate gene transfer events.

## 3. Results

### 3.1. Characteristics of Organellar Genomes

In this study, we assembled and annotated the complete circular chloroplast genome and the complete circular mitochondrial genome of *B. hypnoides*. The chloroplast genome of *B. hypnoides* is 139,745 bp in length (Figure 1A), with a nucleotide composition as follows: 33.82% A, 35.52% T, 15.29% G, and 15.37% C, resulting in an overall GC content of 30.66%. It contains 59 protein-coding genes (PCGs), 2 ribosomal RNA genes (rRNAs), and 11 transfer RNA genes (tRNAs). Among these, 31 genes are related to photosynthesis and 32 genes are associated with self-replication (Table 1). Additionally, two genes (*psbJ* and *trnL*) have two copies, and nine genes contain introns. The mitochondrial genome of *B. hypnoides* is 408,555 bp in length (Figure 1B), with a nucleotide composition as follows: 21.00% A, 21.10% T, 29.00% G, and 28.90% C, resulting in an overall GC content of 57.90%. It contains 41 protein-coding genes (PCGs), 3 ribosomal RNA genes (rRNAs), and 18 transfer RNA genes (tRNAs). Among these, 18 genes are involved in oxidative phosphorylation, and 34 genes are associated with self-replication (Table 2). A total of 4 genes (*trnG*, *trnS*, *trnW*, and *trnF*) have two copies and 15 genes contain introns. Compared to the chloroplast, the mitochondrial genome contains more intron-containing genes, with individual genes hosting a higher number of introns (Table 1 and Table 2). The *cox1* gene of mitochondrial genome harbors the largest intron, spanning multiple genes such as *trnQ* and *rns*, which are commonly observed in green plants [31].

To assess the quality of the assembly, we employed two strategies utilizing both Illumina sequencing reads mapping and PCR-based length validation across various regions, with the incorporation of the most phylogenetically related species as a reference. The mapping results revealed that the chloroplast genome of 139,745 bp was fully covered by 850,013 reads, and the mitochondrial genome of 408,555 bp was fully covered by 525,266 reads, with overlapping reads at every site (Figure 1C,D). In the chloroplast genome, the depth ranged from a minimum of 957× to a maximum of 1644×, with an average of 1531.11× (Appendix A). For the mitochondrial genome, the depth varied between a minimum of 426× and a maximum of 1395×, averaging 1105.72× (Appendix A). Neither of the two organellar genomes exhibited regions with extremely low depth. It is noteworthy that the coverage depths mentioned in Section 2.3 of the Materials and Methods refer to the ratio of the total length of reads aligned to the mitochondrial and chloroplast libraries to the total length of the assembly results. However, since the libraries include organellar sequences from all species, overlaps may increase the sequencing depth artificially. Therefore, the reported depth here is lower than that described in Section 2.3. Therefore, we conclude that the two assembled organellar genomes are devoid of gaps. We found that the chloroplast genome is somehow different in length compared with its closest relatives (also named *B. hypnoides*), the results for which were published in 2011 by Lü et al. [32]. We performed whole genome sequence alignment with these two genomes and identified the variant positions 14,797–24,892, 48,370–48,371, 89,147–89,211, and 119,901–120,513. The mitochondrial genome also showed significant divergence from its closest relatives, results from which were published in 2024 by Ochiai et al. [33] (Appendix A). We designed PCR primers upstream and downstream of the variant sites in both the chloroplast and mitochondrial genomes (Appendix A), such that the amplification products would span the variant sites, allowing for the verification of the assembly accuracy. The lengths of the PCR products confirmed the complete correctness of our assembly results. We hypothesize that these differences between our assembly and the relative species arise from geographical isolation, rather than being artifacts of the assembly process (Figure 1E).

### 3.2. Repeat Sequence

Repeat sequences mediate homologous recombination, leading to inversions and translocations, which are essential in the structural variation of organellar genomes [34]. To explore the underlying mechanisms responsible for the differences in genome size and genomic rearrangements among organellar genomes across various species, we analyzed the profile of repeat sequences (mainly simple sequence repeats [SSRs] and dispersed repeats) in our organellar genomes and their relatives. The analysis of SSRs in chloroplast genomes revealed similar characteristics in *B. hypnoides* compared with its 10 closely related species and 3 outgroup species. The chloroplast genome of *B. hypnoides* comprises 34 SSRs, predominantly mononucleotide repeats (94.12%, all A/T) and a few dinucleotide repeats (5.88%, all AT/AT), with no trinucleotide or higher-order repeats (Figure 2A, Appendix A). This pattern of A/T preference was also observed in the mononucleotide repeats of 10 closely related and 3 outgroup species, where mononucleotide repeats dominate (≥68%) (Appendix A). Among the 14 species, only 6 contain trinucleotide or higher-order repeats. Specifically, *Lambia antarctica*, *Cephaleuros diffusus*, and *Cephaleuros virescens* have only trinucleotide repeats, while *Codium arabicum* and *Codium fragile* have both trinucleotide and tetranucleotide repeats. *Pedobesia claviformis* contains hexanucleotide repeats. Notably, the 3 outgroup species exhibit significantly higher SSR counts and larger genome sizes compared to *B. hypnoides* and its 10 closely related species (Figure 2A).

In the mitochondrial genome, SSRs in *B. hypnoides* and its five closely related species and three outgroup species exhibit completely different characteristics. The mitochondrial genome of *B. hypnoides* contains 128 SSRs, far exceeding the count in the chloroplast genome, and all of them are mononucleotide repeats (Figure 2B, Appendix A). Unlike the chloroplast genome, the mitochondrial genome shows a strong preference for C/G repeats (89.84%) (Appendix A). In the *Bryopsis* and *Ostreobium* genera, along with the three outgroup species, mononucleotide repeats dominate (≥82.76%), but in the three related species of *Caulerpa*, tetra- to hexanucleotide repeats are more abundant (≥62.90%) (Appendix A). In terms of SSR abundance, the two species of *Bryopsis* exhibit significantly higher SSR counts than the other seven species, correlating with the genome size (Figure 2B).

The location of SSRs is similar among the chloroplast and mitochondrial genomes. In the SSRs of the *B. hypnoides* chloroplast genome, 30 (88.24%) are located in intergenic regions, 2 (5.88%) in gene exons, and 2 (5.88%) span intergenic and exon regions (Appendix A). In the mitochondrial genome, 106 (82.81%) SSRs are in intergenic regions, 2 (1.56%) in exons, and 20 (15.62%) in introns (Appendix A), indicating that SSRs are predominantly located in intergenic regions.

The analysis of dispersed repeats showed that only the mitochondrial genomes of *Caulerpa lentillifera* and *Ulva flexuosa* contain reverse repeats (R), while other species (except for *Pedobesia claviformis*, which lacked dispersed repeats ≥55 bp) contain only forward (F) and palindromic (P) repeats (Figure 2C,D). However, the number and type preferences of dispersed repeats varied greatly among the organellar genomes of different species. In chloroplast genomes, the three outgroup species exhibited significantly higher numbers of dispersed repeats than other species (Figure 2C). Across all species, organellar genomes showed a preference for shorter dispersed repeats (Appendix A).

### 3.3. Phylogenetic Analysis

Using 50 common chloroplast protein-coding genes, we reconstructed a maximum likelihood (ML) phylogenetic tree for 14 species. Three *Cephaleuros* species were used as the outgroup. The six *Bryopsis* and *Lambia* species formed one clade, while the five *Pedobesia* and *Codium* species formed another (Figure 3A). The four *Bryopsis* species were found to be closely related to *B. hypnoides*. In comparison to three species of *Codium* and *Pedobesia*, *Lambia* was more closely related to *Bryopsis*, forming a distinct cluster, while *Codium* and *Pedobesia* were placed in another cluster. The support values from Neighbor-Joining, Bayesian, and ML methods were robust enough to validate this phylogenetic relationship.

Using seven common mitochondrial protein-coding genes, we reconstructed an ML phylogenetic tree for nine species. Three *Ulva* species served as the outgroup. The three *Bryopsis* and *Ostreobium* species formed one clade, while the three *Caulerpa* species formed another (Figure 3D). *Bryopsis* sp. KO-2023 was found to be the closest relative to *B. hypnoides*, while three *Bryopsis* species and *Ostreobium* species formed one cluster, and three *Caulerpa* species formed another cluster. Phylogenetic relationships were strongly supported by the Neighbor-Joining, Bayesian, and ML methods. Collinearity analysis showed strong conservation among the mitochondrial genomes of *Bryopsis plumosa* and five closely related species, as well as among the three outgroup species. However, there was significant divergence between closely related species and outgroup species, with evidence of inversions (Figure 3D).

Due to the limited number of complete organellar genome sequences available for the order Bryopsidales, the phylogenetic tree constructed using 10 chloroplast genomes and 5 mitochondrial genomes revealed the evolutionary relationships. While our data provide valuable insights into the phylogeny and evolutionary processes within Bryopsidales, a more comprehensive study will require further data support.

An additional indicator of species diversification within an evolutionary framework is the conservatism of gene complement. Among 14 chloroplast genomes, the 50 shared genes constitute a substantial portion of each species’ genome, ranging from 57.47% to 86.21% (Appendix A). Within specific genera, such as *Bryopsis*, *Lambia*, *Pedobesia*, *Codium*, and *Cephaleuros*, the number of common genes is 58, 81, 77, 75, and 61, respectively. Notably, *Lambia* and *Pedobesia* each possess four unique genus-specific common genes, accounting for 4.94% and 5.19% of their genus-level common genes, respectively, whereas *Bryopsis*, *Codium*, and *Cephaleuros* do not exhibit any (Figure 3B). Among the five *Bryopsis* species, all 58 genes are shared, indicating that all genes in *B. hypnoides* are genus-level common genes. Additionally, *Bryopsis* sp. KO-2023, *Bryopsis* sp. HV04063, and *B. plumosa* have 4, 7, and 12 unique genes, respectively (Figure 3C).

In contrast to chloroplast genomes, the nine mitochondrial genomes share only seven genes, which make up a smaller proportion of each species’ genome, ranging from 31.82% to 14.89% (Appendix A). Within the genera *Bryopsis*, *Ostreobium*, *Caulerpa*, and *Ulva*, the number of common genes is 40, 54, 8, and 29, respectively. *Bryopsis*, *Caulerpa*, and *Ostreobium* have 9, 1, and 20 unique genus-specific common genes, respectively, while *Ulva* lacks any. Among the two *Bryopsis* species, all 40 genes are shared, suggesting that all genes in *B. hypnoides* are genus-level shared genes, with *Bryopsis* sp. KO-2023 having 14 unique genes (Figure 3F).

### 3.4. Gene Rearrangment

Genome synteny analyses indicate the structure conservation of relative species genomes and, thus, are generally used to give evidence for evolution. From the chloroplast genome synteny maps (Figure 3A), we observe significant synteny among the chloroplast genomes of multiple species, particularly within the genus *Codium*, indicating that they have maintained similar gene order arrangements during evolution. Within the genus *Bryopsis*, the chloroplast genomes of different species exhibit a certain degree of synteny, with two species of *B. hypnoides* maintaining a high level of synteny. This suggests that they occupy relatively close positions on the evolutionary tree and may have undergone similar evolutionary trajectories. Despite the presence of synteny, there are still some gene rearrangements among different *Bryopsis* species, including gene insertions, deletions, and inversions. Compared to *Bryopsis*, species from other genera, such as the three species within *Cephaleuros*, exhibit more pronounced gene rearrangements in their chloroplast genomes, potentially reflecting their greater evolutionary distance or exposure to different evolutionary pressures. Notably, the three *Cephaleuros* species have significantly larger genome sizes than other genera. Furthermore, the substantial expansion of the chloroplast genome length in *Cephaleuros* is likely attributed to the vast number of repeats present in its genome, approximately 124 times that of Bryopsidales. In summary, the chloroplast evolutionary rate of the *Bryopsis* genus is inferred to be notably slower than that of the outgroups.

In contrast, the mitochondrial genome synteny reveals substantial and large sequence rearrangements within the *Bryopsis* genus. In comparison, the mitochondrial synteny of the outgroup genus *Ulva* is highly conserved. It is worth noting that the two *Bryopsis* species have significantly larger genome sizes than other genera, and the positions of intergenic regions are not conserved between the two species. Conversely, the three *Ulva* species, which belong to the outgroup, exhibit compact gene arrangements and significantly smaller mitochondrial genomes. In conclusion, the mitochondrial evolutionary rate of the *Bryopsis* genus is notably faster than that of the outgroups, exhibiting an opposite trend compared to the chloroplast.

### 3.5. Codon Usage Bias

The genome of the chloroplast in *B. hypnoides* comprises 12,932 codons. Among these, 1360 codons are dedicated to encoding leucine (Leu), constituting the highest frequency at 10.52% of the total. Conversely, only 146 codons are responsible for encoding cysteine (Cys), representing the lowest frequency, with just 1.13% (Appendix A). Relative Synonymous Codon Usage (RSCU) refers to the ratio of the actual usage frequency of a codon to the expected usage frequency in the absence of any bias. It directly reflects the codon’s preference. If the RSCU > 1, the actual usage frequency of the codon is higher than other synonymous codons; if the RSCU = 1, the usage frequency is equal to that of other synonymous codons [35]. In our results, 28 codons exhibited RSCU values greater than 1, suggesting a higher-than-average actual usage frequency. Conversely, 34 codons had RSCU values below 1, indicating a lower usage frequency. Within the Leu codons, TTA possessed the highest RSCU value at 3.98, whereas CTG had the lowest at 0.10 (Figure 4A).

The mitochondrial genome encompasses 11,900 codons (Appendix A). Within this genome, leucine (Leu) holds the highest frequency with 1257 codons (10.56%), whereas methionine (Met) has the lowest frequency with 253 codons (2.13%), as noted in Appendix A. An RSCU analysis revealed that 31 codons exhibit RSCU values above 1, signifying higher usage, while another 31 codons have RSCU values below 1, indicating lower usage. Specifically, among Leu codons, TTA boasts the highest RSCU value of 1.93, whereas CTC displays the lowest at 0.55 (Figure 4B).

Both the chloroplast and mitochondrial genomes of *B. hypnoides* demonstrate a preference for codons terminating in A/T. However, the chloroplast genome exhibits a more pronounced codon usage bias compared to the mitochondrial genome. With the exception of Met and Trp, which possess a single codon each, the proportion of A/T-ending codons for other amino acids ranges from 75.19% to 93.00% in the chloroplast genome and from 50.38% to 79.60% in the mitochondrial genome, as outlined in Appendix A.

### 3.6. Gene Transfer Between the Two Organelles

Gene transfer from the chloroplast to the mitochondrion is a common occurrence during the long-term evolutionary process of plants [36]. Mitochondrial genomes typically contain sequences derived from plastid DNA, referred to as mitochondrial plastid DNAs (MTPTs) [37,38]. In this study, we analyzed the homologous sequences between the mitochondrial and chloroplast genomes of *B. hypnoides*, marking the first such report within the genus *Bryopsis*. In the two organellar genomes of *B. hypnoides*, we identified 16 homologous sequences ranging from 40 to 666 bp in length (total length: 2573 bp), accounting for approximately 1.84% and 0.63% of the chloroplast and mitochondrial genome lengths, respectively (Appendix A). In the chloroplast genome, 14 sequences (87.50%) were located in gene regions, including 8 (50.00%) in rRNA genes and 2 (12.50%) in intergenic regions. In the mitochondrial genome, 15 sequences (93.75%) were located in gene regions, with 1 (6.25%) in an intergenic region (Figure 5, Appendix A).

## 4. Discussion

Compared to chloroplast assembly, the correct and complete assembly of plant mitochondrial genomes is more challenging. As a result, the number of complete plant mitochondrial genomes in the NCBI database is much lower than that of complete chloroplast genomes, and the number of species with both organellar genomes is even lower. Despite significant advancements in sequencing and assembly methods in recent years, leading to a notable increase in the number of complete chloroplast genomes, the number of mitochondrial genomes remains low [39]. This is primarily due to the presence of long repeat sequences, mitochondrial plastid DNAs (MTPTs), and nuclear mitochondrial transfer DNAs (NUMTs). In addition, the lack and imperfection of tools specifically designed for mitochondrial genome assembly and annotation is also a major contributing factor to this outcome. We simultaneously sequenced and assembled the complete chloroplast and mitochondrial genomes of *B. hypnoides*, providing data support and a reference for the study of the genus *Bryopsis*.

Compared to the nuclear genome, the majority of plastid and mitochondrial genomes are much smaller. The typical length of plant plastid genomes ranges from 115 kb to 165 kb, while mitochondrial genome lengths vary widely, from 66 kb to 12 Mb. In this study, the chloroplast and mitochondrial genomes of *B. hypnoides* we assembled were 139,745 bp and 408,555 bp in length, respectively, which aligns with the typical sizes of plant organellar genomes.

Chloroplasts and mitochondria are the only two organelles with endosymbiotic origins. In recent decades, advances in sequencing technologies have led to a rapid increase in the number of published organellar genomes, revealing significant evolutionary divergence in terms of gene content [39]. During their endosymbiotic evolution, both organelles underwent substantial genome reduction, retaining only a few essential functional genes (many of the lost genes have been transferred to the nuclear genome [40,41,42,43,44], which is a major reason for the stark differences in gene content between these two organellar genomes). However, key processes, such as photosynthesis and respiratory electron transport chains, as well as partial protein synthesis machinery, remain largely intact.

Genes such as NdhA and NdhC, which are commonly lost in plant chloroplast genomes [45], are also absent in the *B. hypnoides* chloroplast genome. Additionally, the NADH-dehydrogenase genes are completely absent from the *B. hypnoides* chloroplast genome. One of the most stable genes in chloroplast genomes, the Rpl20 gene, which is often used as a molecular marker for plastids, was also found in the *B. hypnoides* chloroplast genome. In contrast, mitochondrial genomes of most plants contain far fewer functional genes than chloroplast genomes, yet their total length is much larger [9,46]. The ancestral mitochondrial genome lost many protein-coding genes in different lineages [47], and significant differences in tRNA gene content can even be observed within the same genus [48,49]. Furthermore, in the *B. hypnoides* mitochondrial genome, we identified a large number of open reading frames (ORFs) of unknown function. Previous studies have confirmed that upstream ORFs (uORFs) play a crucial role in regulating plant metabolism, development, disease resistance, and nutrient absorption [50]. However, whether functional uORFs exist in plant organellar genomes and how they function requires further investigation.

Repeat sequences, including simple sequence repeats and dispersed repeats, play significant roles in biological evolution, gene regulation, and genome stability. Simple sequence repeats are widely distributed across organellar genomes and can influence gene expression regulation and serve as molecular markers. Repeats longer than 50 bp can mediate recombination [51], and the likelihood of recombination is directly related to the repeat’s length, with smaller repeats (50–100 bp) rarely causing recombination [52]. Among all organellar genomes we analyzed, we did not find any large repeats (>1000 bp), but a few medium-sized repeats (100–1000 bp) and numerous small repeats (50–100 bp) were identified. Therefore, the conservation of organellar genomes in these species may partly be attributed to the absence of homologous recombination mediated by large repeats. Compared to the chloroplast genome (about 50%), the proportion of coding regions in plant mitochondrial genomes is significantly lower, ranging from 10% to 1% [53]. In other words, mitochondrial genomes consist largely of non-coding DNA, a substantial portion of which is repeat sequences.

The taxon Chlorophyta, which comprises the bulk of green algae, exhibits an extraordinarily wide variety of body structures [54]. Species in *Bryopsis* can develop thalli that reach over 30 cm in height, featuring characteristic side branches. Intriguingly, these thalli lack cell walls to separate their numerous nuclei, resulting in a ‘coenocytic’ structure [2]. This unique feature has led to a lot of research in evolutionary biology [1,2]. Despite the widespread lack of organellar genome data for green algae in public databases, we have observed correlations between phenotypes and evolution from the limited but available genome sequences of existing species. Our findings indicate that species with closer phylogenetic relationships exhibit more similar phenotypes. For instance, the coenocytic genus *Bryopsis* and the mononucleate genus *Ulva* can be distinguished based on mitochondrial phylogenetic trees. It is our belief that, with the release of more organellar genome data and ecological data in the future, the significant evolutionary role under the adaption for *Bryopsis* and other green algae will ultimately be unveiled.

The organellar genomes of *B. hypnoides* show a strong preference for codons, with a marked tendency toward codons ending in A/T, although the chloroplast genome exhibits a stronger preference than the mitochondrial genome. The study of codon usage patterns is important not only for further genetic research on the species, but also for transgenic research. If the codon preference of an exogenous gene differs from that of the recipient genome, it may lead to methylation, potentially reducing the expression of the transgene or causing silencing [55]. Thus, our study provides an essential foundation for future transgenic research on *B. hypnoides*.

The extensive rearrangements observed in the mitochondrial genome may be attributed to intracellular gene transfer (IGT) [56]. Thus, we speculate that the fragmentation of the mitochondrial genome could be due to the integration of fragments from the chloroplast genome. Studies suggest that IGT can form homologous regions within mitochondrial genes [57]. Currently, there is no unified model for IGT events in plants, as they may occur as independent, random events [10]. The increasing number of published nuclear and organellar genome sequences in recent years has provided the opportunity to study the patterns and mechanisms of IGT across broader taxonomic levels. New tools have been developed for studying IGT [58], which will lay an important foundation for understanding its contribution to the shaping of genetic diversity within plant genomes and across species.

## 5. Conclusions

In conclusion, our study has achieved a significant milestone by successfully assembling the chloroplast and mitochondrial genomes of *B. hypnoides* for the first time, with respective sizes of 139,745 bp and 408,555 bp. Comparative genomic analysis has unveiled a remarkable conservation of repetitive sequences and structural integrity within the organelle genomes of species belonging to the order Bryopsidales. Furthermore, phylogenetic analysis, utilizing common genes from both chloroplast and mitochondrial genomes, has illuminated the evolutionary relationships among Bryopsidales species. Notably, within the species *B. hypnoides*, the mitochondrial and chloroplast genomes each exhibit their own specific and pronounced codon usage biases. Our observations hint at extensive gene transfer between these genomes, potentially indicative of intracellular gene transfer mechanisms. These findings contribute invaluable organellar genome data to the knowledge base of the Bryopsidales order and establish a foundational reference for future genetic evolution and phylogenetic investigations.

## Figures and Tables

**Figure 1 biomolecules-15-00278-f001:**
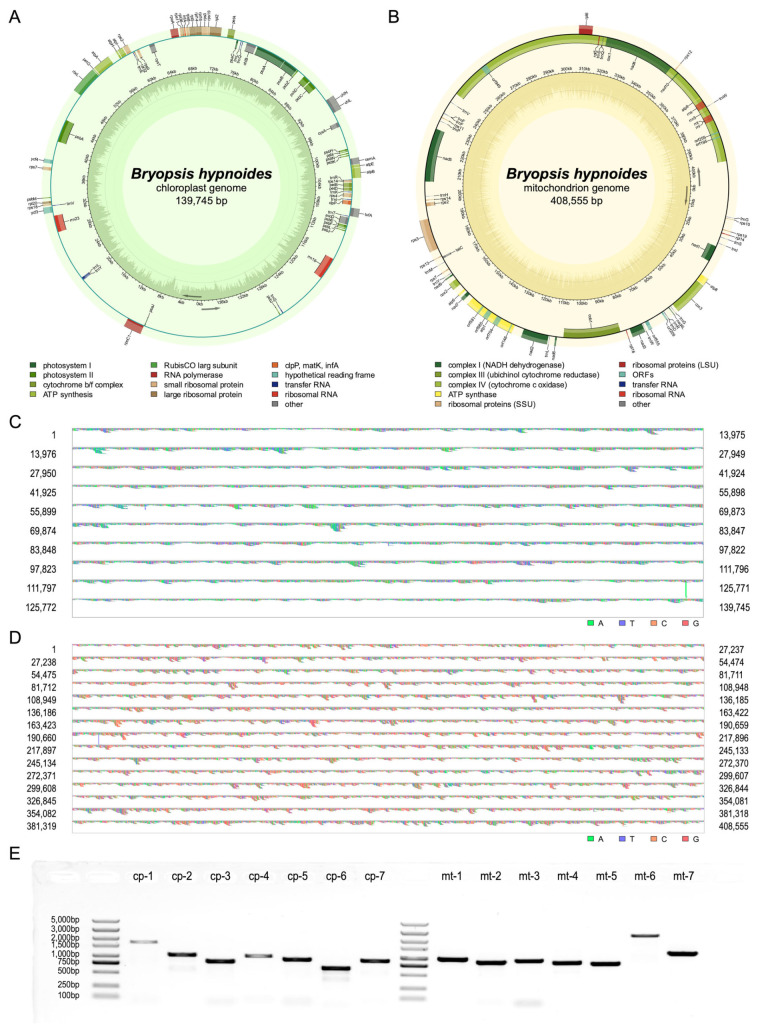
Information of organellar genomes’ assembly and annotation of *Bryopsis hypnoides*. (**A**,**B**) Circular genome map of the chloroplast (**A**) and mitochondrial (**B**) genomes of *B. hypnoides*. The genomic characteristics transcribed in the clockwise and counterclockwise directions are depicted on the inner and outer regions of the circular map, respectively. The functional classification of genes in the organellar genomes is visually represented using color-coding. The GC content is visually represented in the inner circle using a dark green (or yellow) plot. The GC content of each gene is indicated within the gene by dark areas. (**C**,**D**) Results of sequencing reads mapped to the assembled chloroplast (**C**) and mitochondrial (**D**) genomes. Different colors represent different bases. (**E**) PCR results for 14 loci on the organellar genomes. Columns 1 and 9 are markers. Original images can be found in Appendix A.

**Figure 2 biomolecules-15-00278-f002:**
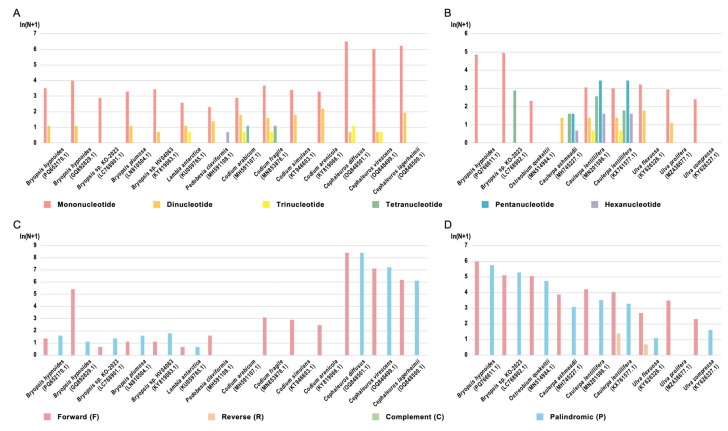
Repeat sequence of *B. hypnoides* organellar genomes. (**A**,**B**) The number of SSRs in the chloroplast (**A**) and mitochondrial (**B**) genomes of *B. hypnoides*. Different colors represent repeat units of different lengths. (**C**,**D**) The number of dispersed repeat sequences in the chloroplast (**C**) and mitochondrial (**D**) genomes of *B. hypnoides*. Different colors represent repeat of different types. ln(N + 1) is the ordinate, while N is the number of repeats of each type.

**Figure 3 biomolecules-15-00278-f003:**
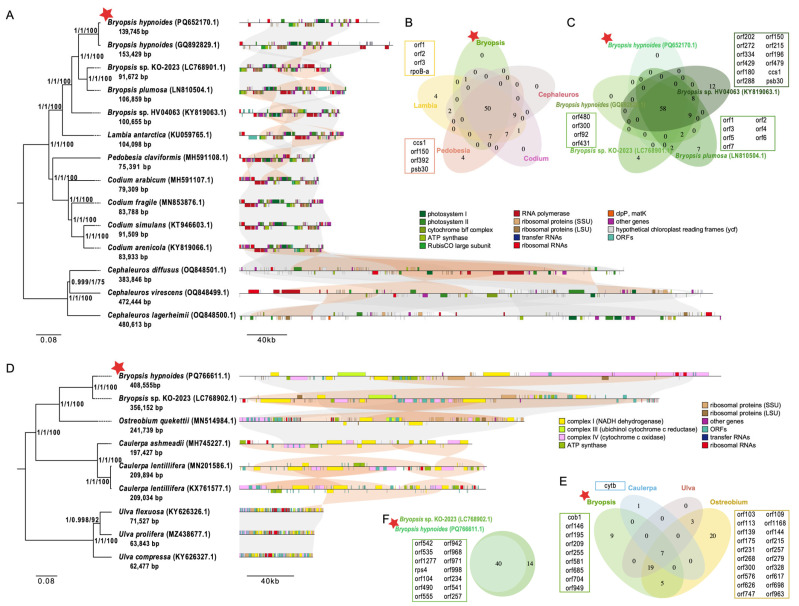
Phylogenetic analysis in order Bryopsidales. (**A**,**D**) Phylogenetic tree based on 50 or 7 homologous protein-coding genes in 14 or 9 plants’ chloroplasts (**A**) and mitochondrial (**D**) using maximum likelihood (ML) analysis. Numbers in the nodes are support values of Neighbor-Joining, Mrbayes test, and the ML bootstrap from 1000 replicates. Collinearity of protein-coding sequences between organellar genomes of *B. hypnoides* and other species showed on the left of the tree, and conserved blocks are visualized in gray (forward) or orange (inversions). The red star represents *B. hypnoides*. (**B**,**E**) The distribution of the genus-level common genes between different genera. Genes in boxes are genus-level common genes of each genus. The red star represents genus *Bryopsis*. (**C**,**F**) The distribution of the species-level common genes in order Bryopsidales. Genes in boxes are species-level common genes of each species. The red star represents *B. hypnoides*.

**Figure 4 biomolecules-15-00278-f004:**
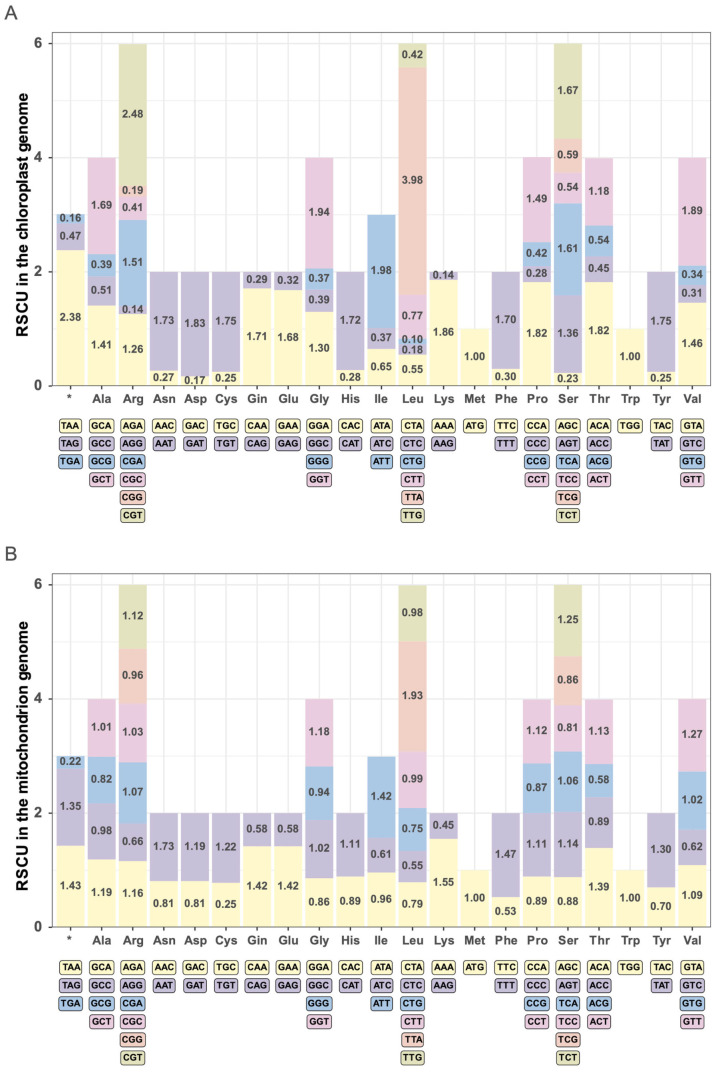
Relative Synonymous Codon Usage (RSCU) analysis in *B. hypnoides* chloroplast (**A**) and mitochondrial (**B**) genomes. The abscission represents the different amino acids, the corresponding codon of each amino acid (distinguished by different colors) is shown below the abscission, and the ordinate is the RSCU value.

**Figure 5 biomolecules-15-00278-f005:**
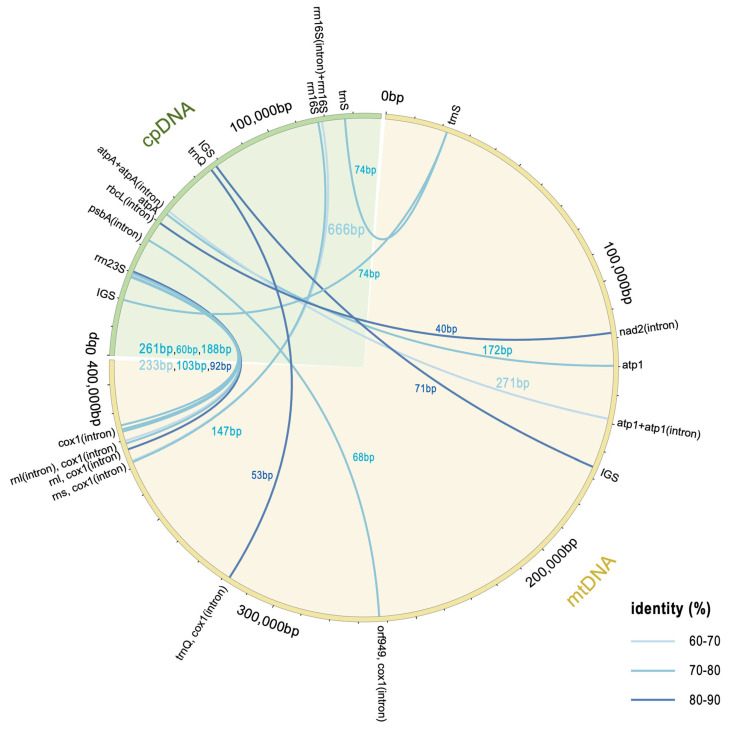
Gene transfer sequence between organellar genomes. The chloroplast genome is shown as a green arc and the mitochondrial genome as a yellow arc. The positions of the transferred genes are connected by lines; the thickness of the line represents the length of the transferred fragment, and the color represents the identity value of the fragment.

**Table 1 biomolecules-15-00278-t001:** Gene functional classification and composition of the *Bryopsis hypnoides* chloroplast genome.

Category	Gene Group	Gene Name
Photosynthesis	Subunits of photosystem I	*psaJ*, *psaC*, *psaA* *, *psaB*
	Subunits of photosystem II	*psbM*, *psbA* *, *psbZ*, *psbD*, *psbC*, *psbH*, *psbI*, *psbN*, *psbK*, *psbE*, *psbF*, *psbL*, *psbJ*(2)
	Subunits of cytochrome b/f complex	*petG*, *petA*, *petB*, *petD*
	Subunits of ATP synthase	*atpA* **, *atpH*, *atpI*, *atpE*, *atpB*
	Large subunit of rubisco	*rbcL* *
	Subunits photochlorophyllide reductase	*chlB*, *chlN*, *chlL*
Self-replication	Proteins of large ribosomal subunit	*rpl20*, *rpl32*, *rpl36*, *rpl5* *, *rpl14*, *rpl16*, *rpl2* *
	Proteins of small ribosomal subunit	*rps18*, *rps7*, *rps2*, *rps9*, *rps11*, *rps8*, *rps3*, *rps19*, *rps14*, *rps4*
	Subunits of RNA polymerase	*rpoC1*, *rpoA*
	Ribosomal RNAs	*rrn23S*, *rrn16S* **
	Transfer RNAs	*trnY*, *trnL* *(2), *trnV*, *trnE*, *tr*nQ, *trnR*, *trnK*, *trnI*, *trnT*, *trnS*
Other genes	Protease	*clpP*
	Envelope membrane protein	*cemA*
	Acetyl-CoA carboxylase	*accD*
	Translation initiation factor	*infA*
	other	*cysT*, *cysA*, *tufA*
Unknown	Conserved open reading frames	*ycf3* *, *ycf4*

Note: * gene with only one intron; ** gene with two introns; gene (2): gene with two copies.

**Table 2 biomolecules-15-00278-t002:** Gene functional classification and composition of the *Bryopsis hypnoides* mitochondrion genome.

Category	Gene Group	Gene Name
Oxidative phosphorylation	Subunits of complex I (NADH dehydrogenase)	*nad1* *, *nad4L*, *nad3* *, *nad6*, *nad2* *^2^, *nad7*, *nad9*, *nad5* *^7^, *nad4* *^6^, *nad10*
	Subunits of complex III (ubichinol cytochrome reductase)	*cob1* *^6^
	Subunits of complex IV (cytochrome c oxidase)	*cox3* *^3^, *cox2* *, *cox1* *^15^
	Subunits of ATP synthase	*atp8* *, *atp1* *^5^, *atp6*, *atp9*
Self-replication	Proteins of large ribosomal subunit	*rpl14*, *rpl16*, *rpl6* *, *rpl5*
	Proteins of small ribosomal subunit	*rps10*, *rps19*, *rps7*, *rps13*, *rps3* *^3^, *rps2*, *rps14*, *rps11*, *rps12*
	Ribosomal RNAs	*rns* *, *rrn5*, *rnl* *^2^
	Transfer RNAs	*trnG*(2), *trnS*(2), *trnI*, *trnR*, *trnD*, *trnW*(2), *trnL*, *trnY*, *trnM*, *trnH*, *trnP*, *trnF*(2), *trnV*, *trnQ*
Other genes	Twin-arginine translocase	*tatC*
Unknown	Conserved open reading frames	*orf209*, *orf555*, *orf146*, *orf704*, *orf685*, *orf581*, *orf949*, *orf255*, *orf195*

Note: * gene with only one intron; *^n^ gene with n introns; gene (2): gene with two copies.

## Data Availability

The assembled genome and annotation are also available from NCBI with GenBank accession ID: PQ652170 (chloroplast) and PQ766611 (mitochondria).

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
