# Peer review of "Comparative Genomics of Bryopsis hypnoides: Structural Conservation and Gene Transfer Between Chloroplast and Mitochondrial Genomes"

_biomolecules, 2025, doi:10.3390/biom15020278_

Round 1

Reviewer 1 Report

Comments and Suggestions for Authors

Dear Authors,

Thank you for submitting your manuscript titled "Comparative Genomics of Bryopsis hypnoides: Structural Conservation and Gene Transfer between Chloroplast and Mitochondrial Genomes." After reviewing your manuscript, I found it to be a comprehensive and well-structured study, offering valuable insights into the organellar genome characteristics and evolutionary patterns of B. hypnoides. Your work is a commendable contribution to the field.

There are only a few minor revisions needed for clarity and precision. Please consider the following suggestions:

  1. Abstract (Line 21-22):

    • Current: "Based on the data, we conducted genetic comparison involved the repeat sequences, phylogenetic relationships, codon usage preferences, and gene transfer between the two organellar genomes."
    • Suggested revision: "Based on the data, we conducted a genetic comparison involving repeat sequences, phylogenetic relationships, codon usage preferences, and gene transfer between the two organellar genomes."
  2. Introduction (Line 34-35):

    • Current: "Bryopsis hypnoides, a unicellular and multinucleate green alga belonging to the genus Bryopsis, is widely distributed from tropical to polar oceans, typically growing on intertidal and subtidal rocks or in rock pools."
    • Suggested revision: "Bryopsis hypnoides, a unicellular and multinucleate green alga of the genus Bryopsis, is widely distributed from tropical to polar oceans, typically growing on intertidal and subtidal rocks or in rock pools."
  3. Materials and Methods (Line 107-109):

    • Current: "To ensure the accuracy of genome assembly, sequencing reads were mapped back to the assembled genomes using Samtools v1.21[20], and depth coverage and gap regions were visualized with Tablet v1.21.02.08[21]."
    • Suggested revision: "To ensure the accuracy of genome assembly, sequencing reads were mapped back to the assembled genomes using Samtools v1.21[20], and depth coverage along with gap regions were visualized with Tablet v1.21.02.08[21]."
  4. Results (Line 153-156):

    • Current: "The chloroplast genome of B. hypnoides is 139,745 bp in length (Figure 1A), with nucleotide composition as follows: 33.82% A, 35.52% T, 15.29% G, and 15.37% C, resulting in an overall GC content of 30.66%. It contains 59 protein-coding genes (PCGs), 2 ribosomal RNA genes (rRNAs), and 11 transfer RNA genes (tRNAs)."
    • Suggested revision: "The chloroplast genome of B. hypnoides is 139,745 bp in length (Figure 1A), with nucleotide composition as follows: 33.82% A, 35.52% T, 15.29% G, and 15.37% C, resulting in an overall GC content of 30.66%. It contains 59 protein-coding genes (PCGs), 2 ribosomal RNA genes (rRNAs), and 11 transfer RNA genes (tRNAs)."
  5. Line 121:
    In your methodology, you used the REPuter tool to identify dispersed repeats. Could you elaborate on how the repeat regions identified in B. hypnoides compare with those in other species' genomes, especially in terms of repeat density or types (palindromic, complementary)?

  6. Line 135-136:
    You used BLASTP for sequence alignment to perform synteny analysis. Could you provide more details on the criteria for selecting conserved blocks? Were these blocks validated by other methods, such as PCR or experimental verification?

These are minor issues that do not affect the overall quality of your manuscript. Once these revisions are made, I believe your manuscript will be ready for acceptance.

Comments on the Quality of English Language

While the manuscript presents excellent research, there are minor grammatical and language clarity issues that could be enhanced by reviewing the text with a native English speaker. This would help in improving the flow and readability of the manuscript.

Reviewer 2 Report

Comments and Suggestions for Authors

This study successfully assembled and analyzed the complete chloroplast and mitochondrial genomes (408,555 bp) of Bryopsis hypnoides, uncovering codon usage biases, distinct repeat sequence patterns, and evidence of gene transfer between the organelles. Several minor issues need to be addressed:

1- The mitochondrial genome (PQ766611) is not accessible, making it challenging to validate the results presented in the study.

2- Although the authors mention sequencing coverage depths (1933× for the chloroplast and 1551× for the mitochondrial genome), they do not explain how these thresholds were determined to ensure high confidence in the assembly. Was a specific cutoff used to define sufficient coverage? Additionally, was coverage uniform across the genome, or were there regions with lower coverage? The study asserts that no gaps were observed in the assemblies, but the exact criteria or tools used for this validation are not described in detail. For example, did the authors assess read mapping efficiency, and if so, what were the specific thresholds?

3- The discussion section provides valuable insights into the comparative genomics and evolutionary dynamics of Bryopsis hypnoides. However, it could be strengthened by connecting these findings to broader ecological or evolutionary contexts.

4- Visualization quality could be improved to enhance clarity and accessibility for a general scientific audience.
